# *Magnaporthe oryzae* effector MoSPAB1 directly activates rice *Bsr-d1* expression to facilitate pathogenesis

Ziwei Zhu [1,2,5], Jun Xiong [1,5], Hao Shi [1,5], Yuchen Liu[1,5], Junjie Yin[1], Kaiwei He [1], Tianyu Zhou [1], Liting Xu [1], Xiaobo Zhu [1], Xiang Lu [1], Yongyan Tang[1], Li Song[1], Qingqing Hou[1], Qing Xiong [1], Long Wang[1], Daihua Ye [1], Tuo Qi[3], Lijuan Zou[3], Guobang Li [1], Changhui Sun[1], Zhiyue Wu[4], Peili Li[4], Jiali Liu[1], Yu Bi[1], Yihua Yang[1], Chunxian Jiang[4], Jing Fan [1], Guoshu Gong[4], Min He [1], Jing Wang[1], Xuewei Chen [1] ✉ & Weitao Li [1] ✉

Fungal pathogens typically use secreted effector proteins to suppress host immune activators to facilitate invasion. However, there is rarely evidence supporting the idea that fungal secretory proteins contribute to pathogenesis by transactivating host genes that suppress defense. We previously found that pathogen *Magnaporthe oryzae* induces rice *Bsr-d1* to facilitate infection and hypothesized that a fungal effector mediates this induction. Here, we report that MoSPAB1 secreted by *M. oryzae* directly binds to the *Bsr-d1* promoter to induce its expression, facilitating pathogenesis. Amino acids 103-123 of MoSPAB1 are required for its binding to the *Bsr-d1* promoter. Both MoSPAB1 and rice MYBS1 compete for binding to the *Bsr-d1* promoter to regulate *Bsr-d1* expression. Furthermore, MoSPAB1 homologues are highly conserved among fungi. In particular, *Colletotrichum fructicola* CfSPAB1 and *Colletotrichum sublineola* CsSPAB1 activate kiwifruit *AcBsr-d1* and sorghum *SbBsr-d1* respectively, to facilitate pathogenesis. Taken together, our findings reveal a conserved module that may be widely utilized by fungi to enhance pathogenesis.

Movements of macromolecules, including proteins, mRNAs, small RNAs, and metabolites, between cells, tissues, and organs are well-known[1,2]. Cross-species movements of these macromolecules occur, including between microorganisms and plants, microorganisms and animals, animals and plants, and plants and plants[3–5]. For example, *Phytophthora sojae* PsXLP1 protein is secreted into soybean cells and binds to soybean GmGIP1 protein to support its infection[5]. Interspecies mRNA transfer occurs among green peach aphids, dodder parasites, and cucumber host plants[6]. Recently, cross-kingdom

movement of macromolecules has become a hot research topic in host-pathogen interactions.

In host-pathogen interactions, fungal pathogens often employ effectors to suppress plant defense responses to invade plants. For instance, rust fungus *Puccinia striiformis* effector Pst_A23 inhibits the functional transcripts of *TaXa21-H* and *TaWRKY53*, hence suppressing the host immune response to invade wheat[7]. Rust fungus *Melampsora larici* effector Mlp124478 binds plant DNA and modulates transcription to reduce expression of resistant genes to facilitate invasion of

[1]State Key Laboratory of Crop Gene Exploration and Utilization in Southwest China, Rice Research Institute, Sichuan Agricultural University, Chengdu, Sichuan 611130, China. [2]Institute for Advanced Study, Chengdu University, Chengdu, Sichuan 610106, China. [3]Ecological Security and Protection Key Laboratory of Sichuan Province, Mianyang Teachers' College, Mianyang, Sichuan 621000, China. [4]College of Agronomy, Sichuan Agricultural University, Chengdu, Sichuan 611130, China. [5]These authors contributed equally: Ziwei Zhu, Jun Xiong, Hao Shi, and Yuchen Liu. ✉e-mail: xwchen88@sicau.edu.cn; weitao-li@sicau.edu.cn

Populus[8]. To promote *P. sojae* infection of soybean, *P. sojae* effector PsCRN108 suppresses expression of heat shock protein (HSP) genes[9]. Bacterium *Xanthomonas* oryzae transcriptional activator-like (TAL) effectors are transported into host cells to induce expression of host SWEET genes, which is required for its infection[10]. However, no reports showed that fungal effectors activate susceptibility genes or genes encoding negative regulators of resistance to accommodate fungal invaders.

Rice blast disease, caused by *Magnaporthe oryzae* (synonym of *Pyricularia oryzae* Sacc.), leads to devastating yield losses worldwide. *M. oryzae*-rice interaction is a well-known model for studying the relationship between plants and pathogens[11]. The development of a disease and a successful infection depends heavily on *M. oryzae* effectors. For example, *M. oryzae* effector AvrPiz-t physically resembles ROD1 and shares the same ROS-elimination cascade with ROD1 to reduce rice immunity and increase virulence[12]. Additionally, *M. oryzae* effectors MoHTR1 and MoHTR2 zinc-finger proteins reduce host defense by preventing expression of resistance-related genes, such as *OsMYB4*, *OsHPL2*, and *OsWRKY45*[13]. *M. oryzae* Molug4 protein enhances rice blast infection by counteracting host OsAHL1-promoted ethylene gene transcription[14]. However, no effectors or secretory proteins of *M. oryzae* or other fungi have been reported to increase host susceptibility by using transcriptional activators to directly induce expression of host genes that favor pathogen infection.

We previously identified a rice *Bsr-d1* gene which encodes a zinc-finger transcription factor that acts as a brake to slow down host defense response and prevent the host immune system from overrunning by increasing $H_2O_2$ degradation in host cells[15]. However, *M. oryzae* exploits this host immune self-control mechanism by inducing *Bsr-d1* expression to suppress the host immune response to *M. oryzae* infection[15]. On the other hand, a wild rice variety developed a natural *bsr-d1* allele that carries in its promoter a base change (SNP33-G) which inhibits this pathogen-induced *Bsr-d1* expression, thereby conferring broad-spectrum resistance to rice blast[15]. Although we have hypothesized that *M. oryzae* may secret an effector to induce *Bsr-d1* expression[15], this postulated effector has remained elusive. Here, we present the discovery of a *M. oryzae* secretory protein associated with *Bsr-d1* (MoSPAB1). MoSPAB1 directly binds to the *Bsr-d1* promoter and induces its expression. MoSPAB1 competes against rice MYBS1 to control *Bsr-d1* expression. Additionally, we identified a DNA-binding domain in MoSPAB1. MoSPAB1 and its target *Bsr-d1* are conserved among fungi and their hosts, respectively, suggesting a working module which may be widely used by fungi for pathogenesis.

## Results

### *M. oryzae* MoSPAB1 directly binds to the promoter of *Bsr-d1* and promotes *Bsr-d1* expression

To identify potential *M. oryzae* factors that activate *Bsr-d1*, we first tested chitin by measuring *Bsr-d1* expression after chitin treatment. Chitin had no effects on *Bsr-d1* RNA levels (Supplementary Fig. 1a). This result is consistent with our hypothesis that *Bsr-d1* may be activated by *M. oryzae* secreted protein(s). Therefore, we used the *Bsr-d1* promoter as bait to search for *M. oryzae* secretory proteins associated with *Bsr-d1* (MoSPAB) by pulling down *M. oryzae* proteins. We obtained two *M. oryzae* proteins (MGG_07390 and MGG_08499) with signal peptides that potentially bind to the *Bsr-d1* promoter. We then tested MGG_07390 (MoSPAB1) and MGG_08499 (MoSPAB2) for their ability to bind to the *Bsr-d1* promoter using the yeast one-hybrid assay. The results show that both MoSPAB1 and MoSPAB2 bind to the *Bsr-d1* promoter, leading to the activation of the HIS2 reporter (Fig. 1a and Supplementary Fig. 1b).

To determine whether the MoSPAB1 and MoSPAB2 effectors regulate *Bsr-d1* expression, we used a dual-luciferase (Dual-LUC) system in *Nicotiana benthamiana* to measure *Bsr-d1* promoter-driven expression levels. The *Bsr-d1* promoter was fused to a LUC as the

reporter (*Bsr-d1:LUC*). *MoSPAB1* and *MoSPAB2* were individually used with the reporter to co-transform tobacco epidermal cells. LUC activity measured as light intensity shows that MoSPAB1, but not MoSPAB2, significantly activates *Bsr-d1:LUC* expression (Fig. 1b and Supplementary Fig. 1c). The amino acid sequence of MoSPAB1 from the 1st residue to the 23rd residue encodes a putative signal peptide (amino acids 1–23) based on the prediction by SignalP5.0. Similarly, the same Dual-LUC system using the rice protoplasts showed that addition of MoSPAB1$^{\Delta SP}$ (MoSPAB1 with signal peptide removed) clearly increased LUC reporter expression in rice protoplasts (Fig. 1c). These results suggest that MoSPAB1 promotes *Bsr-d1* expression.

We next conducted a DNA affinity purification (DAP)-quantitative PCR (qPCR) experiment using the MoSPAB1$^{\Delta SP}$ protein to validate their protein-DNA interaction and to narrow down the MoSPAB1-targeted region in the *Bsr-d1* promoter. We expressed MoSPAB1$^{\Delta SP}$ as a His-MoSPAB1$^{\Delta SP}$ fusion protein in *Escherichia coli* and purified it. Nine promoter fragments (F1-F9) were amplified by PCR for DAP-qPCR (Supplementary Fig. 1d). The results showed that F9 (−92 to +1) was pulled down 3-fold more frequently than the control, which contained an empty His vector (Fig. 1d). This result indicates that MoSPAB1 targets *Bsr-d1* at the region near the starting codon ATG.

In order to confirm that MoSPAB1 directly binds to the *Bsr-d1* promoter, we carried out an electrophoresis mobility shift assay (EMSA). For this purpose, we synthesized five probes using biotin-labeled oligonucleotides containing overlapping domains in F9 of *Bsr-d1* (Supplementary Fig. 1d). According to the EMSA results, His-MoSPAB1$^{\Delta SP}$, but not His tag, binds to probes 1–5 (Supplementary Fig. 1e). We found that probes 1–5 of the *Bsr-d1* promoter contain the conserved motif (C/A)(G/C)(C/G)T(T/C)G(C/A)(T/C) by using the MEME software (Fig. 1e and Supplementary Fig. 1f)[16]. To validate this, we included unlabeled and mutant competitors (mutated to 5′-AATCCCATCAAATCATCCAAAAAAAAAT-3′ and 5′-ACTAAAGA-GAGCTGCTGCTACGTACTA-3′) to compete binding to probes 1 and 2 in EMSA. The binding specificity of His-MoSPAB1$^{\Delta SP}$ to probes 1 and 2 was confirmed because wild-type competitors significantly reduced binding to the probes while mutant competitors had little effects (Fig. 1f, g). These results demonstrate that the MoSPAB1 protein binds to the (C/A)(G/C)(C/G)T(T/C)G(C/A)(T/C) motif in the proximal promoter of *Bsr-d1*.

### *M. oryzae* MoSPAB1 is present in rice cytoplasm and nuclei

To investigate whether *MoSPAB1* is expressed in *M. oryzae*, we detected *MoSPAB1* mRNA levels during development and at infection stage. The results showed that *MoSPAB1* is expressed constitutively. *MoSPAB1* RNA level is particularly high at infection stage (Fig. 2a). It indicates that MoSPAB1 plays an important role in rice-*M. oryzae* interaction.

We then used a yeast genetic assay to test this hypothesis since yeast growth on the raffinose medium depends on the secretion of invertase[17]. The N-terminal regions of MoSPAB1 (amino acids 1–23) and Avr1b (amino acids 1–23, serving as a positive control) were individually fused in-frame to yeast invertase cloned in the pSUC2 vector. When transformed with either construct, the invertase mutant yeast strain YTK12 was able to grow on the YPRAA medium and convert 2, 3, 5-triphenyltetrazolium chloride (TTC) to insoluble red-colored tri-phenylformazan (Fig. 2b). YTK12 cells alone and negative control containing the N-terminus of Mg87[18] were unable to grow on YPRAA plates and remained colorless when treated with TTC (Fig. 2b). These results suggest that MoSPAB1 carries a secretory signal peptide and can be secreted from *M. oryzae*.

To determine the subcellular localization of MoSPAB1 in plant cells, we fused MoSPAB1 to a yellow fluorescent protein (YFP) tag and performed a transient analysis in *N. benthamiana*. Both MoSPAB1-YFP and MoSPAB1$^{\Delta SP}$-YFP were found in the nucleus and cytoplasm of cells (Supplementary Fig. 2). A similar result was obtained in rice protoplast cells (Fig. 2c), suggesting that MoSPAB1 is secreted from *M. oryzae* to

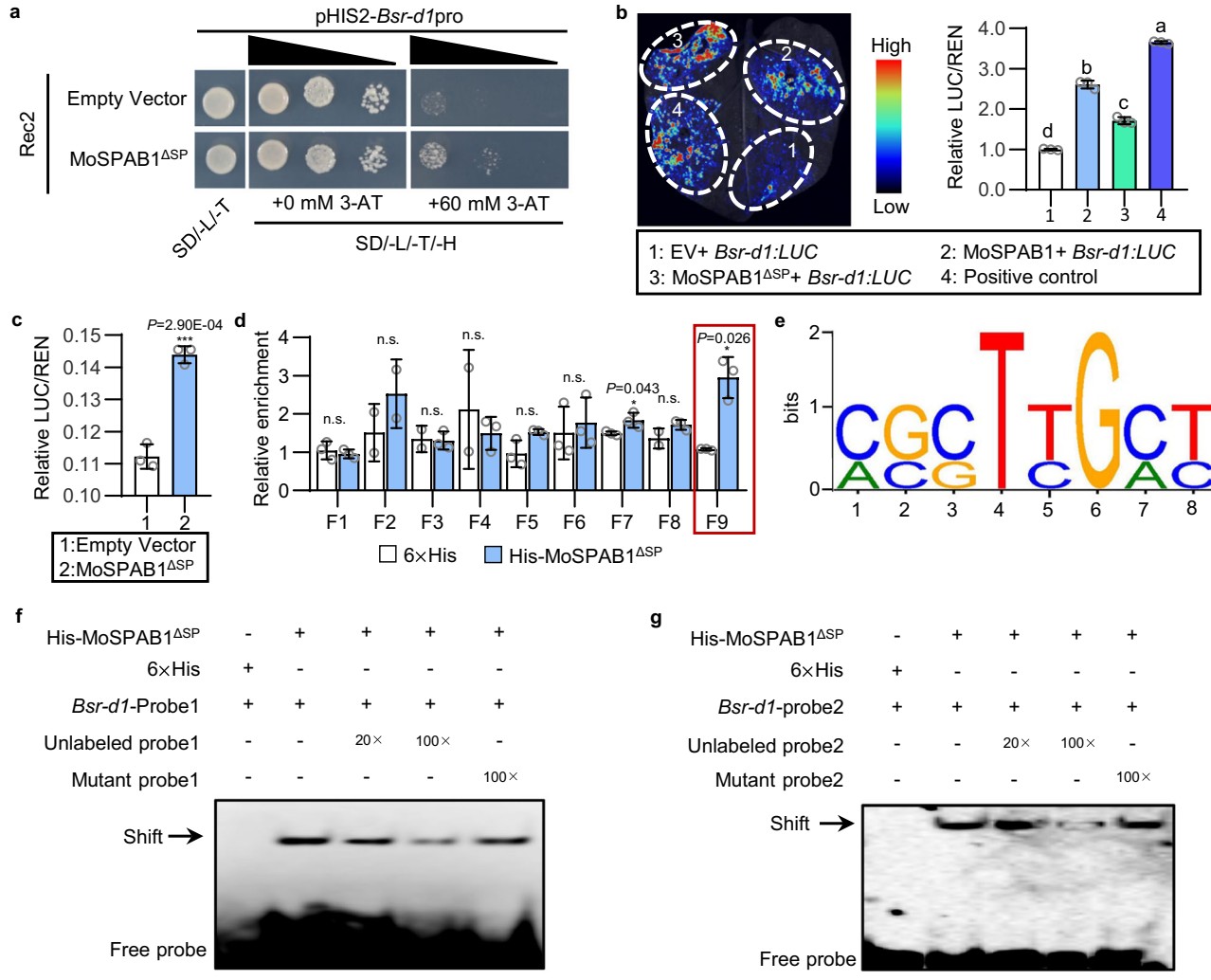

Fig. 1 | *M. oryzae* MoSPAB1 directly binds to the *Bsr-d1* promoter. a Binding of MoSPAB1$^{\Delta SP}$ to the *Bsr-d1* promoter in yeast one-hybrid assay. b Activation of the *Bsr-d1* promoter by MoSPAB1 in luciferase assay using *N. benthamiana*. Luciferase signals were imaged (left panel) and measured (right panel) using a dual-LUC assay 48 hr after infiltration. The positive control contains transcription factor EAT1 and *OsLTP94* promoter driving reporter. EV contains 35 S:YFP as the negative control. firefly luciferase (LUC) activities were normalized to *Renilla* luciferase (REN). Values are mean ± SD, *n* = 3 biologically independent samples. Data analyzed by one-way ANOVA followed by two-sided least significant difference (LSD) test for multiple comparisons. Different letters indicate significant differences (*P* < 0.05). *P*-values are shown in the Source Data file. c Luciferase activity assay for MoSPAB1$^{\Delta SP}$ binding to the *Bsr-d1* promoter in rice protoplasts. The fluorescence signals of LUC and REN

(internal control) were detected after rice protoplasts were co-transfected with the vectors for 16 h. Values are mean ± SD, *n* = 3 biologically independent samples (Student's two-sided *t*-test, ****P* < 0.001). d DNA affinity purification (DAP)-qPCR analysis for enrichment of *Bsr-d1* promoter DNA by recombinant His-MoSPAB1$^{\Delta SP}$ protein. The ratio of His-MoSPAB1$^{\Delta SP}$/6×His ≥ 2 was used as the threshold for significance. Values are mean ± SD, *n* = 3 biologically independent samples (Student's two-sided *t*-test, **P* < 0.05, n.s. indicates no significance). e A conserved sequence bound by MoSPAB1$^{\Delta SP}$ was identified using the MEME suite version 5.4.1. f Binding of His-MoSPAB1$^{\Delta SP}$ to probe 1 of the *Bsr-d1* promoter. Similar results are obtained from three independent biological experiments. g Binding of His-MoSPAB1$^{\Delta SP}$ to probe 2 of the *Bsr-d1* promoter. Similar results are obtained from three independent biological experiments. Source data are provided as a Source Data file.

rice cells and transported into host nuclei. However, MoSPAB1 had no classical nuclear localization signals after sequence analysis. We found that four short regions had at least two K or R amino acids, which might serve as candidate nuclear localization signals (Supplementary Fig. 3a). However, when mutated, each mutant region did not significantly change nuclear localization of MoSPAB1 (Supplementary Fig. 3b). This indicates that MoSPAB1 might use sequences significantly deviating from classical nuclear localization signals or might be facilitated by other factors to enter host nuclei. Furthermore, we fused MoSPAB1 to mCherry to determine its subcellular location in rice cells during *M. oryzae* infection. Fungal transformants expressing MoSPAB1-mCherry driven by its natural promoter were injected into rice sheaths. MoSPAB1-mCherry was not detectable until 36 h post inoculation (hpi). At 36 hpi, only a faint red light was present in the biotrophic interfacial complex (BIC) of *M. oryzae* (Supplementary Fig. 4). At 48 hpi, MoSPAB1-mCherry was clearly visible in the nuclei and cytoplasm

of rice cells (Fig. 2d). The ratio of cytoplasmic MoSPAB1 to nuclear MoSPAB1 was about two (Fig. 2e). These results demonstrate that MoSPAB1 is secreted into host nuclei and suggest that it may play a role in host nuclei.

## *M. oryzae* MoSPAB1 promotes rice susceptibility by regulating *Bsr-d1*

To investigate the role of MoSPAB1 in *M. oryzae* pathogenicity, we generated *MoSPAB1* deletion (Δ*Mospab1*) mutants and their complemented strains containing wild-type *MoSPAB1* (Supplementary Fig. 5). *MoSPAB1* knockout had no effects on spore germination or hyphal growth in *M. oryzae* (Supplementary Fig. 6). When Δ*Mospab1* mutants were used to inoculate rice TP309 (containing *Bsr-d1*) on leaves by punch inoculation, disease lesions were significantly smaller than those caused by wild-type Guy11 or the complemented strain (Fig. 3a). We next examined *M. oryzae* growth on sheath cells of TP309.

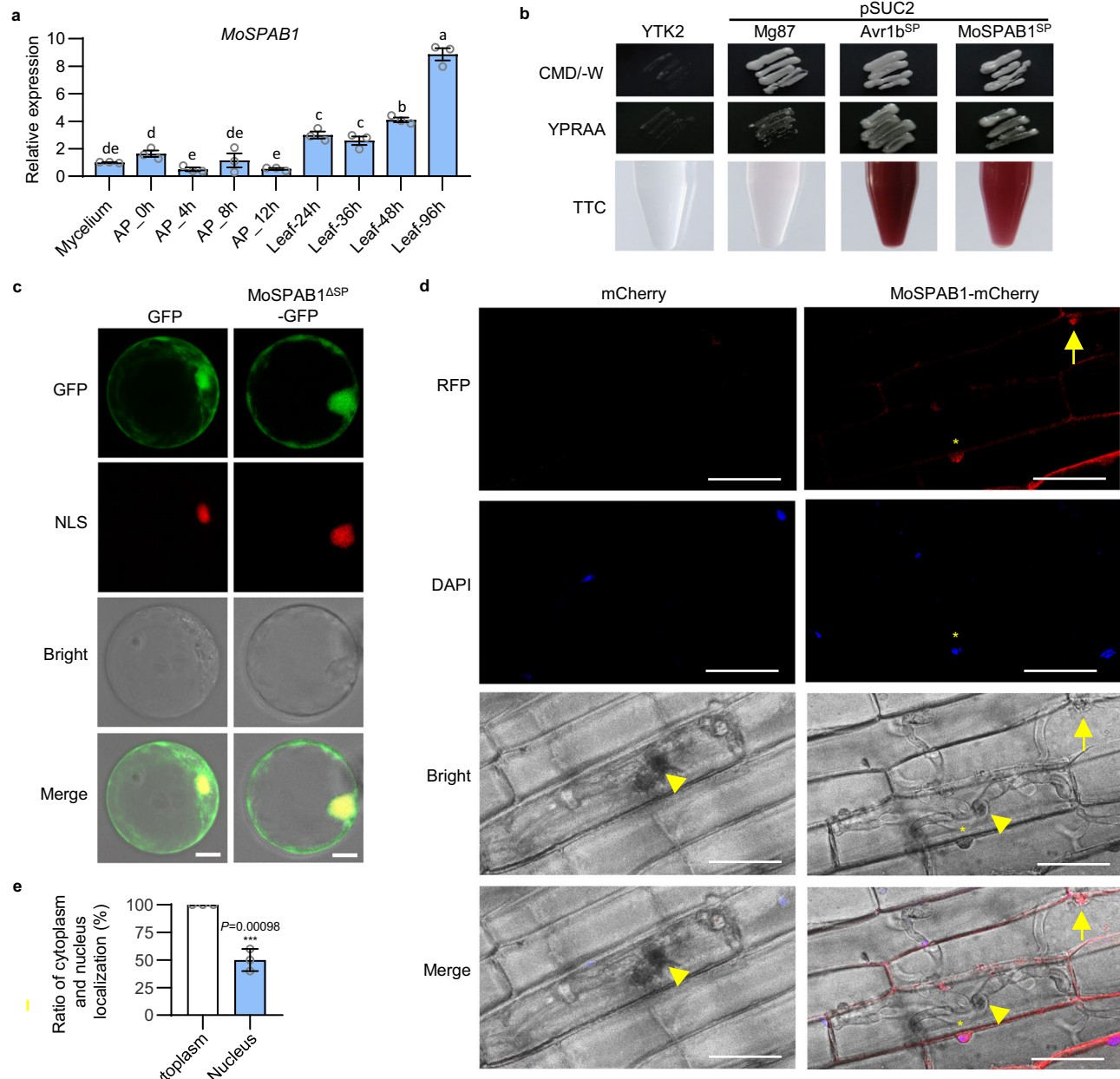

**Fig. 2 | MoSPAB1 is a secreted protein found in the cytoplasm and nuclei of rice cells. a** Expression profile of *MoSPAB1* in Guy11 at various developmental stages. The tubulin gene (*Tub*) was used as an internal control. AP, appressorium; Leaf, leaf after *M. oryzae* infection. Values are mean ± SD, *n* = 3 biologically independent samples. Data are analyzed by one-way ANOVA followed by two-sided least significant difference (LSD) test for multiple comparisons. Different letters indicate significant differences (*P* < 0.05). *P*-values are shown in the Source Data file. **b** Functional validation of the signal peptide of MoSPAB1 using a yeast invertase secretion assay. With a functional signal peptide added, yeast cells secreting invertase grow on both CMD-W and YPRAA plates. Furthermore, invertase enzymatic activity was further tested based on the reduction of TTC to insoluble red-colored TPF. Avr1bSP and Mg87 were positive and negative controls, respectively. **c** Subcellular localization of MoSPAB1^ΔSP in rice protoplasts. MoSPAB1^ΔSP-GFP was expressed transiently in rice protoplasts for 16 h. NLS serves as a nuclear marker indicated by the PBI221-H2B-mCherry. Scale bars, 10 μm. Similar results are obtained from three independent biological experiments. **d** Localization of MoSPAB1-mCherry in rice sheath cells 48 h post inoculation. The upper panel shows fungal cells labeled with mCherry using the *MoSPAB1* promoter to express the protein. DAPI was used to stain the nuclei. Yellow asterisks indicate the presence of MoSPAB1-mCherry in the rice nucleus. Yellow arrowheads indicate the biotrophic interfacial complex (BIC) of *M. oryzae*. Yellow triangles represent the appressorium. Scale bars, 20 μm. Similar results are obtained from two independent biological experiments. **e** The ratio of cytoplasmic and nuclear MoSPAB1. Values are mean ± SD, *n* = 10 biologically independent samples (Student's two-sided *t*-test, ***P* < 0.001). Source data are provided as a Source Data file.

The growth statuses of Guy11 and the complemented strain were more advanced than those of the two Δ*Mospab1* strains at 28 and 40 hpi (Supplementary Fig. 7a, upper panel). Quantitation of the numbers of *M. oryzae* at different stages confirms the results (Supplementary Fig. 7a, lower panel). Meanwhile, Δ*Mospab1* induced more H₂O₂ than wild-type Guy11 and the complemented strain in rice (Fig. 3b).

Furthermore, the Δ*Mospab1* mutant did not induce *Bsr-d1* expression as strongly as wild-type Guy11 or the complemented strain, and its induction subsided significantly faster than wild-type Guy11 and the complemented strain (Fig. 3c). These results suggest that *MoSPAB1* knockout specifically reduces *M. oryzae* pathogenicity to TP309 which harbors wild-type *Bsr-d1*.

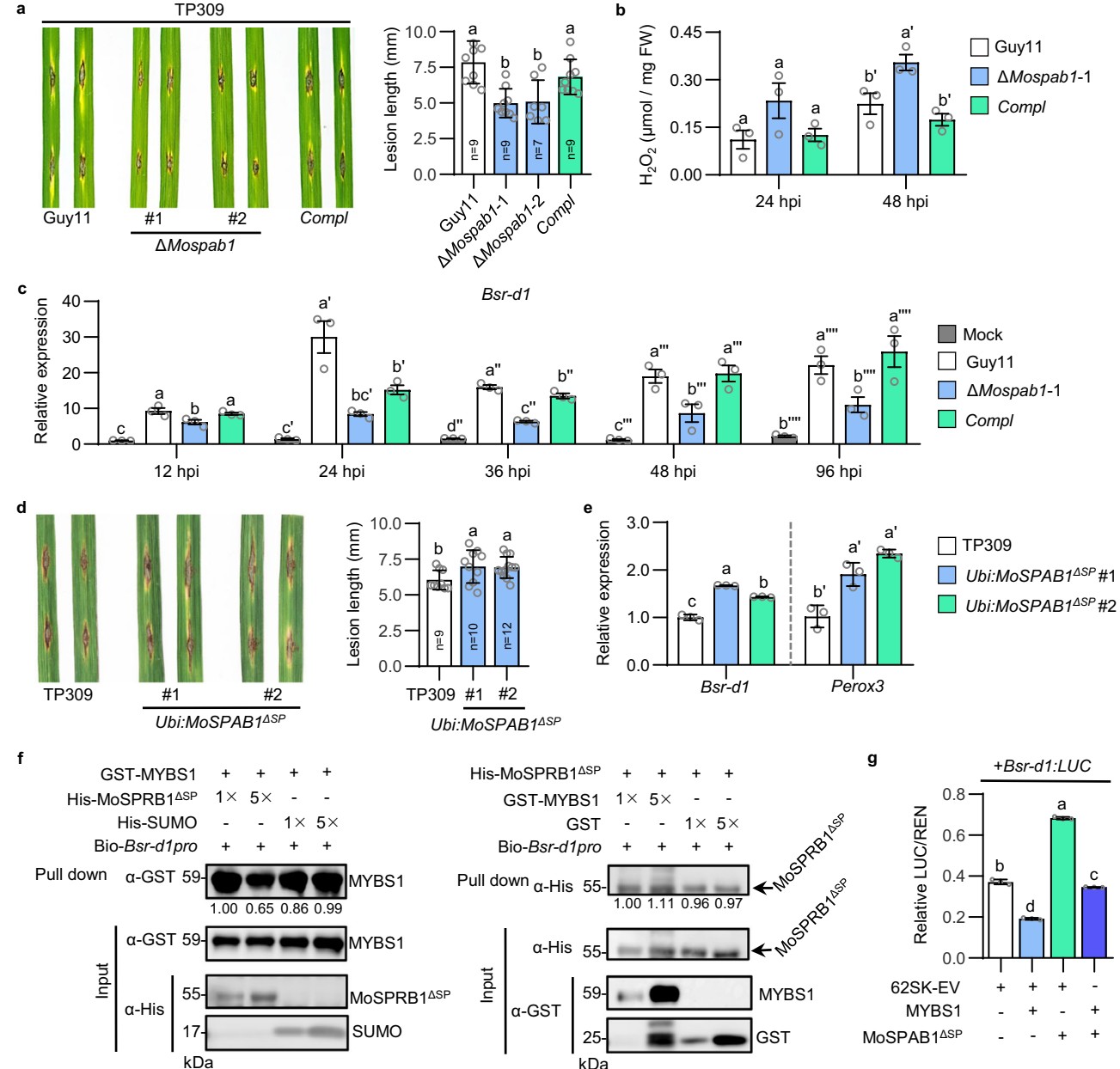

**Fig. 3 | MoSPAB1 promotes rice susceptibility by activating *Bsr-d1*.**
**a** Pathogenicity of Guy11, the Δ*MoSPAB1* mutant, and the complemented strain in rice leaves using punch inoculation. Photos (left panel) were taken and lesion lengths (right panel) were calculated. *n* = 9, 9, 7, and 9, respectively, showed in **a**. **b** H₂O₂ levels in rice after *M. oryzae* inoculation. *n* = 3 biologically independent samples. **c** *Bsr-d1* expression levels in rice leaves after spraying inoculation. RNA samples were extracted from TP309 leaves 12, 24, 36, 48 and 96 hpi with Guy11, the Δ*Mospab1* mutant, and the complemented strain, individually. Mock was inoculated without *M. oryzae*. *n* = 3 biologically independent samples. **d** Rice blast susceptibility with heterologous *MoSPAB1^{ΔSP}* expression. *n* = 9, 10, and 12, respectively, showed in **d**. **e** *Bsr-d1* and *Perox3* expression levels in rice plants with heterologous *MoSPAB1^{ΔSP}* expression. *n* = 3 biologically independent samples. **f** Simultaneous

binding of MoSPAB1^{ΔSP} and MYBS1 to the *Bsr-d1* promoter in a DNA pull-down assay. A 1689-bp biotin-labeled *Bsr-d1* promoter was co-incubated with His-MoSPAB1^{ΔSP} and GST-MYBS1. Increasing concentrations (1×, 5×) of MoSPAB1^{ΔSP} was respectively incubated with a fixed concentration of GST-MYBS1 (left panel), or vice versa (right panel). Intensity of the band was quantified and the ratio was shown below the lane. Similar results are obtained from two independent biological experiments. **g** Effects of MoSPAB1^{ΔSP} and MYBS1 on *Bsr-d1:LUC* reporter expression in a dual-luciferase reporter assay using rice protoplasts. *n* = 3 biologically independent samples. In (**a**, **b**, **c**, **d**, **e–g**) Data are mean ± SD. Different letters indicate significant differences at *P* < 0.05 (one-way ANOVA followed by two-sided least significant difference (LSD) test for multiple-comparisons). *P*-values are shown in the Source Data file. Source data are provided as a Source Data file.

We then generated transgenic rice overexpressing *MoSPAB1^{ΔSP}* using a ubiquitin promoter (*Ubi:MoSPAB1^{ΔSP}*) and inoculated these plants with *M. oryzae* Guy11 (Supplementary Fig. 8). *Ubi:MoSPAB1^{ΔSP}* plants developed disease lesions significantly longer than those on wild-type TP309 plants (Fig. 3d). We next examined *M. oryzae* growth on sheath cells of TP309 and *Ubi:MoSPAB1^{ΔSP}* lines. The growth statuses of Guy11 on *Ubi:MoSPAB1^{ΔSP}* lines were more advanced than those on TP309 at 24 and 36 hpi (Supplementary Fig. 7b, upper panel).

Quantitation of the numbers of *M. oryzae* at different stages confirms the results (Supplementary Fig. 7b, lower panel). This result suggests that *MoSPAB1* enhances host susceptibility to *M. oryzae*.

To test if MoSPAB1 regulates *Bsr-d1* expression in rice, we first detected the expression levels of *Bsr-d1* and BSR-D1 target gene *Perox3* in *Ubi:MoSPAB1^{ΔSP}* transgenic lines. Our result showed that *Bsr-d1* and *Perox3* were significantly up-regulated in the *Ubi:MoSPAB1^{ΔSP}* lines (Fig. 3e). Then, Guy11, the Δ*Mospab1* mutants and its complemented

strain were used to inoculate TP309, TP309 with *Bsr-d1* knockout (*Bsr-d1*KO), and TP309 overexpressing *Bsr-d1* (*Bsr-d1*OE). In comparison to TP309, the *Bsr-d1*KO lines developed smaller lesions, whereas the *Bsr-d1*OE lines developed longer lesions when inoculated with Guy 11 or the complemented strain. However, when the two *Bsr-d1*KO lines were inoculated with the ΔMospab1 mutants, they developed similar lesions as TP309 (Supplementary Fig. 9), indicating that the effect of *Bsr-d1* is tightly linked to MoSPAB1. These results further suggest that MoSPAB1 promotes host susceptibility by up-regulating *Bsr-d1*.

## MoSPAB1 and MYBS1 compete to control *Bsr-d1* expression
MYBS1 binds to the *Bsr-d1* promoter and inhibits *Bsr-d1* expression[15], whereas MoSPAB1 binds to the *Bsr-d1* promoter and promotes *Bsr-d1* expression. To test whether and how MoSPAB1 and MYBS1 compete for *Bsr-d1*, we first tested for possible interaction between MoSPAB1 and MYBS1 using the yeast two-hybrid assay and found that they did not directly interact (Supplementary Fig. 10). This result prompted us to investigate whether MoSPAB1 and MYBS1 compete to bind to the *Bsr-d1* promoter. To test their binding ability, we incubated purified His-MoSPAB1$^{\Delta SP}$ and GST-MYBS1 proteins with biotinylated *Bsr-d1* promoter, pulled down the biotinylated promoter, and detected His-MoSPAB1$^{\Delta SP}$ and GST-MYBS1 with antibodies. With MoSPAB1$^{\Delta SP}$ concentration increased by 5-fold, the GST-MYBS1 amount pulled down slightly decreased to 0.65-fold; with GST-MYBS1 concentration increased by 5-fold, the MoSPAB1$^{\Delta SP}$ amount pulled down barely changed (1.11-fold) (Fig. 3f). These results suggest that, even though binding of MoSPAB1 to the *Bsr-d1* promoter may be slightly stronger than MYBS1, their binding to the *Bsr-d1* promoter are mostly independent.

We then tested the outcome of their simultaneous binding to the promoter. We used the Dual-LUC system in rice protoplast cells to detect the effects of simultaneous binding of MoSPAB1 and MYBS1 to the *Bsr-d1* promoter. *Bsr-d1*:LUC was used as the reporter and *Renilla* luciferase used as the reference. Addition of MoSPAB1 clearly increased LUC expression from the *Bsr-d1* promoter whereas addition of MYBS1 significantly decreased LUC expression. Simultaneous addition of MoSPAB1 and MYBS1 canceled most of the enhancing effect of MoSPAB1 on the *Bsr-d1* promoter (Fig. 3g). These results suggest that *M. oryzae* MoSPAB1 and rice MYBS1 compete to control *Bsr-d1* expression.

## Amino acids 103-123 of MoSPAB1 are required for binding to the *Bsr-d1* promoter
MoSPAB1 has the ability to bind to the *Bsr-d1* promoter. However, it is an expressed protein with no known functional domains. To find the critical domain binding to the (C/A)(G/C)(C/G)T(T/C)G(C/A)T cis-motif, we first analyzed and identified conserved regions of MoSPAB1 using the MEGA X software[19]. We found that MoSPAB1 homologous proteins are present in many diverse fungi, indicating that SPAB1 is a highly conserved protein. We aligned these MoSPAB1 homologous proteins (*e*-value < 5e-90), which revealed four conserved regions (Supplementary Figs. 11 and 12). We then determined which region is required for binding to the (C/A)(G/C)(C/G)T(T/C)G(C/A)(T/C) motif by deleting each conserved region of MoSPAB1. Our results showed that MoSPAB1 lost binding to the *Bsr-d1* promoter when the second conserved region (amino acids 103 to 123) was deleted (Fig. 4a). The sequence of this region is TLIERVEVSLVSHDTTFQVIR, which is predicted to form a domain containing an intact β-sheet, a random coil, and a portion of helix (Fig. 4b, c). The underlined amino acids in this domain are completely conserved (Fig. 4b), indicating their importance in this domain.

## Function of SPAB1 is likely conserved in fungi
To determine if the function of SPAB1 is conserved among diverse fungi, we cloned *MoSPAB1* homologous genes (*CfSPAB1* and *CsSPAB1*)

from *Colletotrichum fructicola* and *Colletotrichum sublineola*, respectively. Furthermore, we also cloned the promoters of *Bsr-d1* homologous genes (*AcBsr-d1* and *SbBsr-d1*) from kiwifruit (*Actinidia chinensis*) and sorghum (*Sorghum bicolor*), the host plants of the two fungi. The conserved motif (C/A)(G/C)(C/G)T(T/C)G(C/A)(T/C) is present in the promoters of *AcBsr-d1* and *SbBsr-d1* (Supplementary Fig. 13). We also used the Dual-LUC system to determine if MoSPAB1 homologous proteins are able to induce expression of the *Bsr-d1* homologous genes. The results showed that CfSPAB1$^{\Delta SP}$ and CsSPAB1$^{\Delta SP}$ significantly activated the *AcBsr-d1* and *SbBsr-d1* promoters, respectively, resulting in significantly higher LUC expression from the promoters (Fig. 4d, e). These results indicate that, like MoSPAB1, CfSPAB1 and CsSPAB1 likely play a role in fungal pathogenicity.

Because previous studies have used *N. benthamiana* as a host to assess *Colletotrichum* pathogenicity[20,21], we infected *N. benthamiana* with *C. fructicola* and *C. sublineola* to investigate potential roles of CfSPAB1 and CsSPAB1 in pathogenicity. Tobacco leaves overexpressing *CsSPAB1*$^{\Delta SP}$ via *Agrobacterium*-mediated infiltration developed larger lesions (2.3-fold) compared to control after *C. sublineola* inoculation (Fig. 4f). Similarly, tobacco overexpressing *CfSPAB1*$^{\Delta SP}$ via *Agrobacterium*-mediated infiltration developed larger lesions (2.7-fold) compared to control after *C. fructicola* inoculation (Fig. 4g). These results suggest that *CfSPAB1* and *CsSPAB1* promote *C. fructicola* and *C. sublineola* infection, respectively, supporting a role for them in fungal pathogenicity.

## Discussion
Here, we present a model to summarize our results (Fig. 5). When *M. oryzae* infects rice, the pathogen secretes MoSPAB1 into host cells where it enters the nucleus. MoSPAB1 then binds to the *Bsr-d1* promoter, promoting *Bsr-d1* expression to facilitate *M. oryzae* infection. BSR-D1 can activate expression of several peroxidase genes (*LOC_Os05g04470*, *LOC_Os10g39170* and *LOC_Os01g73170*) and thus promote $H_2O_2$ degradation by these peroxidases, leading to susceptibility[15,22]. The rice host, on the other hand, employs MYBS1 to inhibit *Bsr-d1* expression to resist *M. oryzae* infection[15].

This discovery represents a mechanism in which the pathogen employs a transcriptional activator (MoSPAB1) to activate a host "immune-brake" gene (*Bsr-d1*) to facilitate its infection while the host employs a competing transcriptional repressor (MYBS1) to repress the same gene to protect itself from invasion. The working module of MoSPAB1-*Bsr-d1* likely exists in other plant-pathogen interactions because MoSPAB1 homologs are present in many other fungal pathogens, including *C. fructicola* and *C. sublineola*, and *Bsr-d1* homologous genomic sequences, including their promoters, are conserved in other host plants, including kiwifruit and sorghum.

## MoSPAB1 represents a strategy that exploits a host "immune-brake" gene for fungal infection
Fungal pathogens typically deliver effectors into host cells to counter the host immune response and facilitate infection[13]. Numerous studies have found that fungal effectors influence plant immunity by interacting with host proteins. However, relatively few reports showed that fungal effectors regulate host genes expression[7,13,23]. To facilitate fungal infection, all of these effectors suppress positive immune regulators of hosts. Conceivably, pathogens can exploit host weakness to attack. As a good example, MoSPAB1 activates the "immune-brake" *Bsr-d1* in rice nuclei to boost host susceptibility (Fig. 3a, c). This demonstrates that the mechanism mediated by MoSPAB1 is distinct from those mediated by previously reported effectors.

In contrast to suppressing or breaking the host defense systems, *M. oryzae* detects the *Bsr-d1* gene as a key point of host weakness and exploits it with MoSPAB1. Thus, in conjunction with other strategies used by *M. oryzae*, the secreted MoSPAB1 protein would synergistically

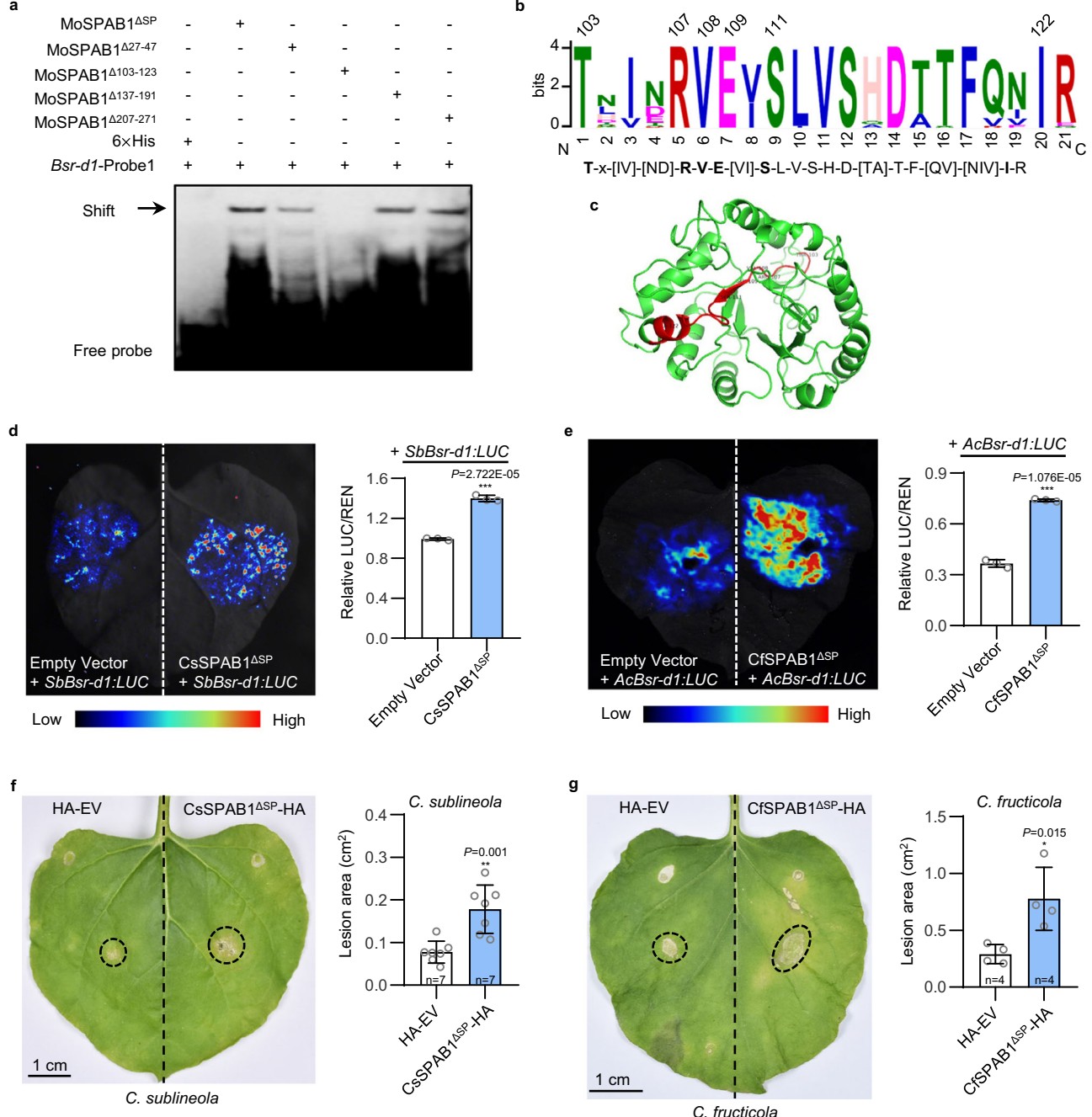

**Fig. 4 | Amino acids 103-123 of MoSPAB1 are required for binding to the *Bsr-d1* promoter. a** Detection of MoSPAB1 domains required for binding to the conserved cis-element using EMSA. The regions of amino acids 27-47, 103-123, 137-191, and 207-271 of MoSPAB1$^{ΔSP}$ were deleted one by one, tagged with 6×His. Similar results are obtained from two independent biological experiments. **b** A LOGO sequence of the critical peptide (aa 103-123) of MoSPAB1. The conserved motif was predicted by aligning 45 homologous proteins using the MEME Suite. **c** A 3-D structure model for MoSPAB1 constructed by I-TASSER. The red portion represents the critical aa 103-123 region. Completely conserved amino acids are labeled. C-score = -1.61. Estimated TM-score = 0.52 ± 0.15. Estimated RMSD = 9.8 ± 4.6 Å. **d** Interaction of CsSPAB1$^{ΔSP}$ from *Colletotrichum sublineola* with the *SbBsr-d1* promoter in luciferase activity assay using *N. benthamiana* leaves. Values are mean ± SD, *n* = 3 biologically

independent samples. **e** Interaction of CfSPAB1$^{ΔSP}$ from *Colletotrichum fructicola* with the *AcBsr-d1* promoter in luciferase assay using *N. benthamiana* leaves. **f** Transient expression of CsSPAB1$^{ΔSP}$ in *N. benthamiana* increases host susceptibility to *C. sublineola*. The pre-wounded *Agrobacterium*-infiltrated leaves were detached and inoculated with *C. sublineola* mycelial plugs after 24 hr post-infiltration. Lesion areas were denoted by circles, and scored 8 days post inoculation. Values are mean ± SD, *n* = 7 biologically independent samples. **g** Transient expression of CfSPAB1$^{ΔSP}$ in *N. benthamiana* increases host susceptibility to *C. fructicola*. Values are mean ± SD, *n* = 4 biologically independent samples. In (**d**–**g**) Data are analyzed by two-sided *t*-test, ***$P$ < 0.001, **$P$ < 0.01, *$P$ < 0.05. Source data are provided as a Source Data file.

improve fungal infection. It is worth noting that MoSPAB1 is also found in the cytoplasm of plant cells (Fig. 2c), indicating that MoSPAB1 might also play a role in the cytoplasm, which would need further investigation.

**Novel fungal secretory proteins with transcriptional activator activity are yet to be fully discovered**

Typically, proteins secreted by fungi do not share significant homology with plant proteins. Similarly, no homologous MoSPAB1 proteins

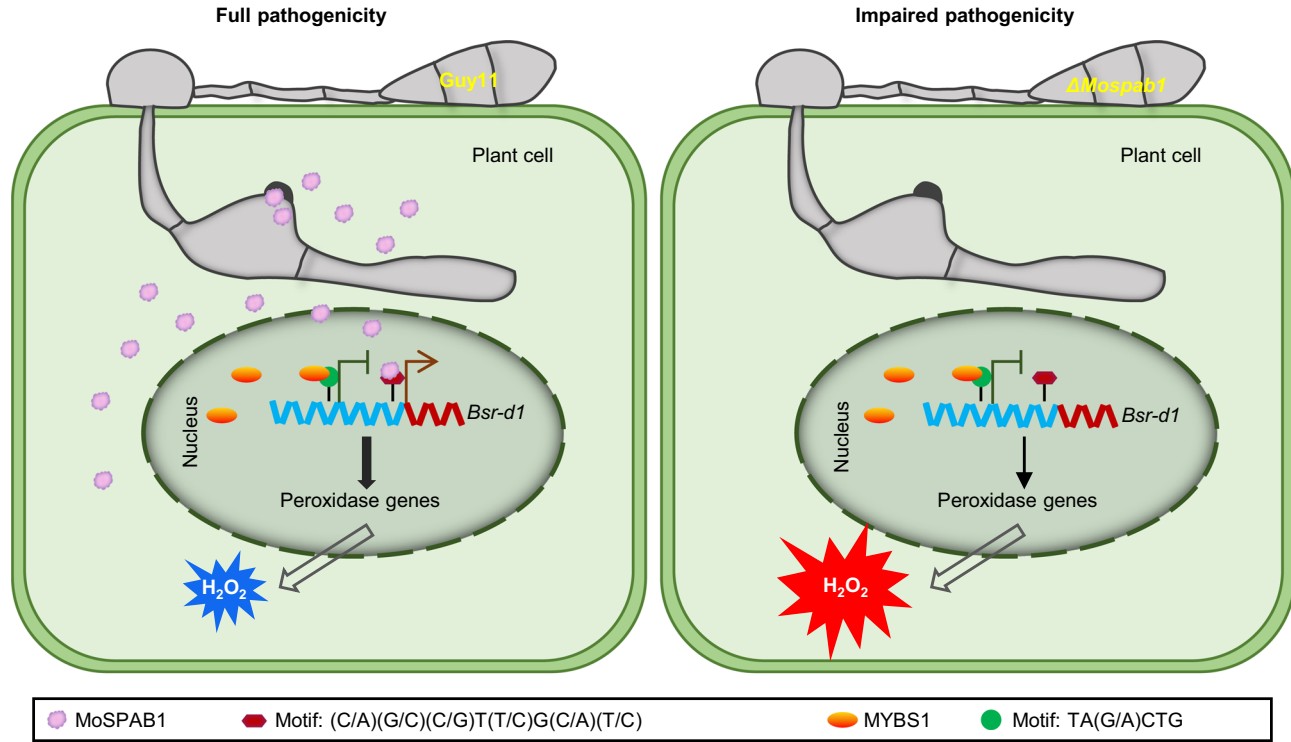

**Fig. 5 | A model for MoSPAB1-mediated pathogenicity.** During *M. oryzae* infection, MoSPAB1 was secreted from invasive hyphae and translocated to rice cytoplasm and the host nucleus. MoSPAB1 binds to the (C/A)(G/C)(C/G)T(T/C)G(C/A)(T/C) motif on the *Bsr-d1* promoter and activates *Bsr-d1*. BSR-D1 accumulation accelerates $H_2O_2$ degradation by inducing expression of peroxidase genes, increasing disease susceptibility. Rice uses MYBS1 to suppress *Bsr-d1* expression in order to combat *M. oryzae* MoSPAB1. Deletion of *MoSPAB1* reduces *M. oryzae* pathogenicity on rice.

were found in plants. Few secretory proteins function as transcription factors, and those that do usually have a known DNA binding domain. For example, *Ustilago maydis* effector Nkd1 has an ethylene-responsive element binding factor-associated amphiphilic repression (EAR) domain[24]. *P. sojae* effector PsCRN108 contains a helix-hairpin-helix (HhH) domain[9]. MoSDT1, MoHTR1, and MoHTR2 all contain a C2H2 zinc-finger domain[13,25]. Pathogen virulence proteins are evolutionarily highly variable, which favors pathogens in the battle between host plants and pathogens. Our study found that MoSPAB1, a secretory fungal protein that lacks a typical domain of known transcription factors, binds to the (C/A)(G/C)(C/G)T(T/C)G(C/A)(T/C) motif in the *Bsr-d1* promoter and activates its expression (Figs. 1 and 3). This shows how a secretory fungal protein without a typical DNA-binding domain may function to regulate host genes during infection.

### The competition between fungal MoSPAB1 and host MYBS1 to control *Bsr-d1* represents a new starting point in pathogen-host interactions

Although several studies reported cases where pathogen proteins compete with plant proteins for interactions with host proteins[26–28], no reports of pathogen proteins competing with plant transcription factors to regulate host genes were available. Here, MoSPAB1 competes against rice MYBS1 for the ability to promote *Bsr-d1* expression although they may simultaneously bind to the *Bsr-d1* promoter (Fig. 3g).

Furthermore, MoSPAB1 appears to bind to the *Bsr-d1* promoter more strongly than MYBS1 (Fig. 3f). This may be due to the multiple tandem MoSPAB1-binding motifs present in the proximal *Bsr-d1* promoter (Supplementary Fig. 1d, f). These tandem motifs give MoSPAB1 a tighter binding to the *Bsr-d1* promoter, which favors *M. oryzae* for successful infection. Our results indicate that a competition between pathogen and host proteins, like the MoSPAB1-MYBS1-*Bsr-d1* model, may be common in pathogen-host interactions.

### The SPAB1-*Bsr-d1* cross-kingdom module is conserved in plant-pathogen interactions

Rice-*M. oryzae* interaction has already become a model for plant-pathogen interactions[11]. Some regulatory modules for plant-pathogen interactions are conserved in the plant immune system. For example, plant pattern recognition receptors (PRRs) FLS2 and EF-TU recognize conserved pathogen-associated molecular patterns (PAMPs) flagellin and elongation factor, respectively[29]. Rice receptor-like kinase OsCERK1 coordinates with CEBiP to initiate the conserved chitin-triggered defense response[30–32]. In this study, we discovered MoSPAB1 homologous proteins in many fungal pathogens, indicating that the MoSPAB1-*Bsr-d1* module may exist widely for cross-kingdom communication between fungi and plants. Especially MoSPAB1 of *M. oryzae* shares high homology with SPAB1s of the *Colletotrichum* genus (Supplementary Fig. 11). Importantly, CsSPAB1 and CfSPAB1 can regulate *SbBsr-d1* in sorghum and *AcBsr-d1* in kiwifruit, respectively (Fig. 4d, e). As a result, we discovered a conserved cross-kingdom regulatory module that may lead to novel strategies to prevent and control anthracnose disease, which is caused by the *Colletotrichum* genus and affects a wide variety of plants[33]. For example, the diseases caused by *Colletotrichum* may be controlled by manipulating expression of these genes. Controlling diseases by targeting the SPAB-*Bsr-d1* module would also differentiate it from those targeting effectors that are directly or indirectly recognized by disease resistance proteins.

## Methods
### Plant varieties and fungal pathogen isolates
Previously reported transgenic lines (Ubi-C1300-*Bsr-d1* and pBGK032-*Bsr-d1*) and wild-type TP309[15] were used in this study. Other plant species include sorghum BTx623 and kiwifruit Hongyang. The homologous gene of *Bsr-d1* in sorghum BTx623 is SORBI_3001G330100 and that in kiwifruit Hongyang is CEY00_Acc21054.

*M. oryzae* isolates, Guy11, two Δ*Mospab1* mutants, and a complemented strain carrying *MoSPAB1*, were used for inoculation. All isolates were grown in a growth chamber at 25°C in a 12 h light/12 h dark photoperiod before being used for inoculation. Other fungi include *Colletotrichum sublineola* str. TX430BB g2-2 and *Colletotrichum fructicola* HX19-3-2. Homologous genes of *MoSPAB1* in *Colletotrichum sublineola* str. TX430BB g2-2 is *CSUB01_11810* and that in *Colletotrichum fructicola* HX19-3-2 is *CGMCC3_g17995*.

## DNA pull-down assay

For screening secretory proteins of *M. oryzae* that bind to the *Bsr-d1* promoter, the *Bsr-d1* promoter was firstly cloned and labeled with biotin. Biotinylated *Bsr-d1* promoter was pre-incubated with dynabeads-streptavidin (Pierce™ Streptavidin Magnetic Beads, Thermofisher). The rice leaves infected by *M. oryzae* were ground into powder in liquid nitrogen and lysed by adding extraction buffer (50 mM Tris-MES, pH 8.0, 0.5 M sucrose, 1 mM $MgCl_2$, 10 mM EDTA, and 1% protease inhibitor cocktail). Then the *Bsr-d1* promoter/strepavidin-dynabeads mixture was incubated with total proteins from infected rice leaves at 25 °C for 4 hr. After magnetic separation, unbound proteins were removed by rinsing with wash buffer (25 mM Tris, 0.15 M NaCl, 0.1% Tween-20, pH 7.2). After SDS-PAGE electrophoresis, isolated DNA binding proteins were collected for LC-MS assay.

To assay competition between MoSPAB1$^{ΔSP}$ and MYBS1 for binding to the *Bsr-d1* promoter, recombinant His-MoSPAB1$^{ΔSP}$ and GST-MYBS1, separately or together, were incubated with biotinylated *Bsr-d1* promoter (1689-bp) at 25 °C for 2 hr. The DNA-protein mixture was incubated in a binding buffer (25 mM Tris, 0.15 M NaCl, pH 7.2). Then the DNA-protein complexes were incubated with dynabeads-streptavidin (Pierce™ Streptavidin Magnetic Beads, Thermofisher) at 25 °C for 1 h. After magnetic separation, unbound proteins were removed by rinsing with wash buffer (25 mM Tris, 0.15 M NaCl, 0.1% Tween-20, pH 7.2). The entire DNA-protein complexes and isolated DNA binding proteins were analyzed by immunoblotting using antibodies against each protein.

## One-hybrid assays in yeast

Full-length cDNAs of two *M. oryzae* genes (*MoSPAB1* and *MoSPAB2*) without signal peptide sequences were amplified and fused in frame with the GAL4 activation domain in pGADT7-Rec2 (Clontech) individually. Then, the fusion construct was used with a reporter vector (*Bsr-d1* promoter-HIS2) to co-transform Y187 yeast cells (Clontech). Sequences of the primers are listed in Supplementary Data 1. Empty vector pGADT7-Rec2 and the *Bsr-d1* promoter-HIS2 plasmid were co-transformed as the control for mating experiments. DNA-protein interactions were determined by growth of the transformants on the nutrient-deficient medium with 60 mM 3-amino-1,2,4-triazole (3-AT, Solarbio), following the manufacturer's manual (Clontech).

## Validation of predicted signal peptide

Functional validation of the predicted signal peptide (SP, SignalP version 5.1) of putative secretory proteins in *M. oryzae* was conducted using a yeast secretion assay described previously[34-36]. The SP coding sequences (amino acids 1–23) of *M. oryzae* candidate secretory protein *MoSPAB1* was amplified and cloned into pSUC2 with in-frame fusion with invertase. The pSUC2-derived plasmid (0.5 μg) was used to transform invertase-deficient yeast strain YTK12. Transformants were selected on yeast minimal tryptophan dropout medium. Then, yeast colonies were plated onto YPRAA plates for invertase secretion assays. At the same time, the invertase enzymatic activity of yeast colonies was detected by the reduction of TTC to insoluble red-colored triphenylformazan.

## Subcellular localization of MoSPAB1 in plants

For subcellular localization of MoSPAB1 in *N. benthamiana*, a cDNA corresponding to the entire coding sequence of *MoSPAB1* was cloned in between *Sac*I and *Kpn*I sites of the pCAMBIA1300-35S-YFP vector to generate pCAMBIA1300-*MoSPAB1* (35 S:MoSPAB1-YFP). *Agrobacteria* carrying the fusion construct ($OD_{600}$ = 0.5) were used to transform or co-transform *N. benthamiana*. RFP-NLS$_{SV40}$[37] and pCAMBIA1300 (35 S:YFP) were used as controls. Fluorescence was examined under a confocal microscope (Leica STELLARIS STED, Germany) 48 h after transformation.

For subcellular localization of MoSPAB1 in rice protoplast cells, a cDNA of *MoSPAB1* without SP was cloned in between the *BamH*I and *Hind*III sites of the pRTVcGFP vector to generate pRTVcGFP-*MoSPAB1*$^{ΔSP}$ (Ubi:MoSPAB1$^{ΔSP}$-GFP). The fusion construct was used to transform or co-transform protoplasts prepared from TP309 seedlings as described previously[38]. PBI221-H2B-mCherry (NLS)[39] and PRTVcGFP (Ubi:GFP) were used as controls. Fluorescence was examined under a confocal microscope 16 h after transformation.

The *MoSPAB1-mCherry M. oryzae* isolate was used to inoculate detached rice sheaths from 4-week-old rice plants as previously described[15]. Briefly, the fungus stably expressing MoSPAB1-mCherry or mCherry was grown on a complete agar medium for 2 weeks for spore production. Spores were collected via flooding of the fungal agar cultures with sterile water, and the spore concentration was adjusted to ~1.0 × 10$^5$ conidia/mL to inoculate detached rice sheaths.

## RNA isolation and reverse transcription-quantitative PCR (RT-qPCR)

RNA isolation and RT-qPCR were conducted as previously described[15]. Briefly, Total RNA was extracted using TRIzol reagent (Invitrogen Life Technologies, Shanghai, China) according to the manufacturer's protocols. cDNA was synthesized using an RNA reverse transcription kit (Vayzme, Nanjing, China). RT-qPCR was conducted using a Bio-Rad CFX96 Real-Time System coupled to a C1000 Thermal Cycler (Bio-Rad, Hercules, CA, USA). The reference gene *ubiquitin 5* (*Ubq5*) was used for normalization of all RT-qPCR data. Sequences of the primers are listed in Supplementary Data 1.

## Electrophoretic mobility shift assay

Light Shift Chemiluminescent EMSA Kit (Beyotime, Shanghai, China) was used in this experiment. Recombinant His-MoSPAB1$^{ΔSP}$, His-MoSPAB1$^{Δ27-47}$, His-MoSPAB1$^{Δ103-123}$, His-MoSPAB1$^{Δ137-191}$, and His-MoSPAB1$^{Δ207-271}$ proteins were all purified using Ni-NTA Sefinose (TM) Resins (Sangon Biotech, Shanghai, China). Biotin was labeled at the 5′-end of *cis*-elements. The biotin-labeled DNA was synthesized by Sangon Biotech (Shanghai, China). The detailed procedure of EMSA follows the manufacturer's instructions. Photos were taken using a charge-coupled device (CCD) camera.

## DNA affinity purification (DAP)-qPCR

Prepared DNA in DAP was applied for qPCR using respective primer pairs (Supplementary Data 1) and a SYBR Green qPCR mix (QuantiNova SYBR Green qPCR master mix Kit, QIAGEN) in a Bio-Rad CFX96 real-time PCR detection system. PCR reactions were conducted as described previously[15]. Expression levels were normalized to the input sample for enrichment detection. The fold enrichment was calculated against the *Ubq5* promoter. A purified 6×His protein served as a negative control.

## Generation of *M. oryzae* gene deletion and complementation

A PCR-based, split-marker deletion method was used for the targeted gene deletion of *MoSPAB1*. The coding region of *MoSPAB1* was targeted for replacement by the selection gene *HPH*, conferring resistance to hygromycin B (Roche). Fungal transformation, selection, and

confirmation of deletion mutants were performed as previously described[40]. For genetic complementation, the *MoSPAB1-mCherry* fusion gene expression vector containing the *MoSPAB1* native promoter was constructed with PCR-directed recombination. Details of the information for all primers, PCR templates, and amplicons were shown in Supplementary Data 1.

### Dual-Luciferase assay
Plasmids for secretory proteins were constructed by cloning full-length cDNAs of *MoSPAB1* and *MoSPAB1*$^{\Delta SP}$ into the pCAMBIA1300 vector under the control of the 35S promoter. The 1689-bp *Bsr-d1* promoter was cloned into the pGreenII-0800 vector, forming a reporter construct. *Renilla* luciferase (REN) driven by the CaMV 35S promoter was used as the internal control. Secretory protein constructs and reporters were used to co-transform tobacco leaf epidermal cells by *Agrobacterium*-mediated infiltration as described previously[41].

For rice protoplasts, the plasmids for secretory proteins were constructed by cloning full-length cDNAs of *MYBS1*, *MoSPAB1*$^{\Delta SP}$, *CfSPAB1*$^{\Delta SP}$, and *CsSPAB1*$^{\Delta SP}$ individually into the pGreenII 62-SK vector under the control of the 35S promoter. This method was also described previously[42]. Diluted D-Luciferin potassium salt (1 mM) (Coolaber, China) was smeared on the tobacco leaf surface, and a live imaging instrument (Bio-Rad ChemiDoc XRS) was used to detect fluorescence. LUC and REN activities were detected using the Dual-Lumi II Luciferase Assay Kit (Beyotime, China) in the GLOMAX96 Microplate Luminometer system (Promega), according to the manufacturer's manual. Preparation and subsequent transfection of rice protoplasts were performed as described previously[38]. Relative LUC/REN ratios were calculated.

### Punch inoculation on rice leaves
Punch inoculation of rice leaves was conducted as previously described[15]. Briefly, the spore concentration of *M. oryzae* in the suspension was adjusted to $1.5 \times 10^5$ conidia/mL before punch inoculation. Procedures are as follows: dip 5 µL spore suspension for each drop using a transfer pipette at two spots on each leaf, keep them in a culture dish that contains 0.1% 6-benzylaminopurine (6-BA) sterile water to keep moist, and measure lesion length using ruler 6 days post-inoculation.

### Two-hybrid assays in yeast
Full-length cDNA sequences of *MYBS1* and *MoSPAB1* without SP were amplified and cloned in pGADT7 and pGBKT7, respectively (Gold Yeast Two-Hybrid, Takara). Empty vectors pGADT7 and pGBKT7 were used as the negative control for mating experiments. pGBKT7-53 and pGADT7-T were used as the positive control. Protein interactions were determined by growth of the transformants on the nutrient-deficient medium, following the manufacturer's manual.

### Determination of reactive oxygen species
A hydrogen peroxide assay kit (Beyotime) was used to measure $H_2O_2$ levels according to the manufacturer's manual. Briefly, 20 mg rice leaves were weighed and 300 µL lysate was added; samples were shaken for 30 s, and then centrifuged at 12000 rpm, 4 °C, for 5 min; the supernatant was pipetted into a 1.5 mL tube, then centrifuged at 12000 rpm, 4 °C, for 5 min; 50 µL sample was placed into a 96-well ELISA detection plate. 100 µL detection solution of hydrogen peroxide was added into each wall, and then mixed and stirred gently. Samples were measured at 560 nm after 30 min at room temperature. The standard curve was y = 158.18x - 16.083, $R^2$ = 0.9995.

### Infection of *N. benthamiana* leaves by *C. sublineola* and *C. fructicola*
*Agrobacterium tumefaciens* cells containing 35S:3 × HA, 35S:CsSPAB1$^{\Delta SP}$-3 × HA, and 35S:CfSPAB1$^{\Delta SP}$-3 × HA ($OD_{600}$ = 0.5) were separately injected into *N. benthamiana* leaves. Leaves were cut off after 24 h. The mycelia of *C. sublineola* and *C. fructicola* were cultured on PDA for 8 days. The mycelial plugs were prepared with a perforator, and placed on scratched sites of detached leaves. The leaves were scratched with pipette tips before use. The 0.1% 6-BA and 1% Huawuque nutrient solution (CHE JETER Technology Co.Ltd, Beijing, China) with pH 7.0 were used to keep moisture and the leaves green. The samples were placed in an incubator at 25 °C. After about 8 days, the leaves with lesions were photographed. Lesion areas were analyzed with Image J.

### Statistics and reproducibility
Leica LAS X Hardware Configurator Version 2020.6.0 was used to collect the micrographs. Image Lab version 3.0 build 11 was used for DNA electrophoresis and WB data collection. MEGA X was used for sequence alignment. MEME Suite 5.4.1 was used for analyzing conserved motifs. SignalP 5.0 was used for analyzing the signal peptide. All data analysis was performed using GraphPad 8.0. The statistical analyses were performed using SPSS 26.0. All values are presented with mean ± SD and the number ($n$) of samples is indicated in the legend or figure. Statistically significant differences between control and experimental groups were determined by one-way ANOVA with two-sided least significant difference (LSD) or Dunnett T3 multiple-comparison test or $t$-test. Differences were considered statistically significant when $P$-value is <0.05. All experiments were repeated at least twice and multiple biological replicates were used in all experiments. No data were excluded from the analyses. The investigators were not blinded to allocation during experiments and outcome assessment.

### Reporting summary
Further information on research design is available in the Nature Portfolio Reporting Summary linked to this article.

## Data availability
The sequence data used in this study are available in the EndemblFungi database under accession code MGG_07390 and MGG_08499 [https://fungi.ensembl.org/Magnaporthe_oryzae/Info/Index], in the National Center for Biotechnology Information database under accession code XP_002465089.1 (SORBI_3001G330100), KDN66937 (CSUB01_11810), and XP_031875374.1 (CGMCC3_g17995) [https://www.ncbi.nlm.nih.gov/], in the Rice Genome Annotation Project database under accession code LOC_Os03g32230, LOC_Os01g34060, and LOC_Os01g73170 [http://rice.uga.edu/], and in the Gramene database under accession code CEY00_Acc21054 [https://www.gramene.org/]. The original data points in graphs and uncropped gel and immunoblotting images are provided in the Source Data files. Source data are provided with this paper.

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

## Acknowledgements

This work was supported by the National Natural Science Foundation of China (31825022, 32121003) to X.C., (31972254 and 32172419) to W.L., 32072407 to X.Z., 32072041 to J.Y., 32001887 to L.W.; National Key Research and Development program (2021YFA1300702), Major Projects in Agricultural Biological Breeding (2022ZD04002), New Cornerstone Science Foundation through the XPLORER PRIZE, New Cornerstone Investigator Program and Major Program of National Natural Science Foundation of Sichuan (2023NSFSC0005) to X.C.; Fok Ying Tung Education Foundation (171021), Innovative Training Program of SAU (202001) and Sichuan National Science Foundation Innovation Research Group from Sichuan Science and Technology Program (2023NSFSC1996), and the Innovative Training Programme of Sichuan Agricultural University (202110626030) to W.L.; National Key Research and Development Program of China for Young Scientists (2022YFD1401400) and the State Key Laboratory of Crop Gene Exploration and Utilization in Southwest China, SAU (SKL-ZY202215) to X.Z.; Sichuan Scientific and Technological Project (2021YFYZ0021) to C.J.; National Natural Science Foundation of Sichuan

(2022NSFSC1622) to J.Y. Thank Dr. Mawsheng Chern for providing critical reading and editing.

## Author contributions

X.C. and W.L. conceived and designed the experiments. Z.Z., J.X., H.S., and T.Z. performed experiments on phenotypic and biochemical assays. Z.Z., J.X., J.Y., Y.L., and L.X. worked on the transgenic lines. Z.Z., J.X., Y.L., and T.Z. performed the experiments on protein localization. Z.Z., J.X., H.S., K.H., C.S., Y.L., and L.X. performed plasmids construction. M.H., J.W., J.Y., X.Z., Y.T., T.Q., J.L., Y.B., Y.Y., G.G., Z.W., P.L., C.J., J.F., and G.L. contributed to reagents, plant and fungal materials. Z.Z., J.X., H.S., X.L., L.S., Q.H., Q.X., L.W., D.Y., and L.Z. collected the data. Z.Z., W.L., and X.C. wrote the manuscript.

## Competing interests

The authors declare no competing interests.
