## [Peer Review File · Nature Communications]

REVIEWER COMMENTS

Reviewer #1 (Remarks to the Author):

General comment:

This study assessed the role of the MoSPAB1 protein of *M. oryzae* in the directly activation of rice Bsr-d1 expression to facilitate pathogenesis. The authors previously identified the rice protein Bsr-d1 that encodes a zinc-finger transcription factor that acts as a brake to slow down host defense response and prevent the host immune system from overrunning by increasing H₂O₂ degradation in plant cells. Through protein pulldowns (using the Bsr-d1 promoter as a bait) the author identified two secreted proteins (MoSPAB1 and MoSPAB2) that binds to Bsr-d1 promoter to facilitate it activation and Bsr-d1 gene expression. The author employed a broad range of molecular techniques to prove their hypothesis. Live cell imaging of fluorescently labeled MoSPAB1 proved it nuclear localization in rice and tobacco cells. Both deletion and overexpression of MoSPAB1 proved that this protein is involved in the virulence of *M. oryzae*. The data analysis and statistics presented in this manuscript are appropriate. However, the authors must show individual data points in all graphs. The manuscript is well written, and the outcomes should serve as references for fungal geneticists and plant pathologists. I support the publication of this manuscript in Nature Communications after proper corrections.

I have some comments that the authors should address.

General Comment: I strongly suggest that the authors should use the term effector for the MoSPAB1 and MoSPAB2. If the author cannot classify these proteins as effectors, they need a strong justification stated in the manuscript.

Specific comments:

Abstract: Fungal pathogens typically use secreted proteins to suppress host immune activators to facilitate invasion. It should be written: secreted effector proteins

Introduction:

Line 71-72: The rice blast disease caused by the *Magnaporthe oryzae* fungus can lead to devastating yield losses worldwide. Please improve the sentence. Rice blast disease, caused by *Magnaporthe oryzae* (synonym of *Pyricularia oryzae* Sacc.), causes devastating yield losses worldwide.

Figure 1: Authors must show individual data points in the graphs of this figure and in all graphs of the manuscript. Please present the graphs in color. Use color-blind friendly colors for the figures. Why the only one bar in Fig1B is in dark gray? Graph legends are touching the axes labels.

Figure 2: Part of the scale bars in the protoplast pictures are outside of the panels. Please improve it. The legend mentions the yellow triangles. It is supposed to be written arrowheads that indicate the location of the BIC. Yellow triangles arrowheads represent indicates the biotrophic interfacial complex (BIC) of *M. oryzae*.

Figure 3: Please present the graphs in color. Panel C must improve resolution.

Figure 4: The authors could make the circles around the lesion in Fig4F and G with a darker color. White is hard to see.

Figure 5: Legend: "During *M. oryzae* infection, MoSPAB1 was secreted from invasive hyphae to rice cytoplasm and the host nucleus". It is missing the word "translocated" It should be written: "During *M. oryzae* infection, MoSPAB1 was secreted from invasive hyphae and translocated to rice cytoplasm and the host nucleus.

References: Please double check journal abbreviations.

Supplemental Figures 9 and 10. The figures have poor resolution. Dendrograms and alignments should be presented in color.

Reviewer #2 (Remarks to the Author):

The authors report a novel mechanism for fungal infection in plant using the model pathosystem, rice and *Magnaporthe oryzae* interactions. The fungal effector MoSPAB1 secreted by the blast fungus binds to the promotor of host gene, *Bsr-d1*, and activates its expression. Higher expression of *Bsr-d1* makes rice plant more susceptible against the blast fungal infection. Furthermore, authors found MoSPAB1 and host MYBS1 compete to bind to the promotor of *Bsr-d1*. Experiments were straight forward and performed scientifically, but the data and figures presented in the current form need to be upgraded or

replaced. And Discussion should be shortened and restructured. Followings are the major and minor comments to improve the manuscript.

1. Localization of MoSPAB1. Fig 2D is not clear to see the localization of MoSPAB1. It is very difficult to figure out where is the appressorium, BIC and rice nucleus and whether they are in invaded or neighboring cells. It should be upgraded or replaced! And what is the ratio of cytoplasm and nucleus localization of MoSPAB1?
2. Expression of MoSPAB1. Its expression was highest at 4 dpi (equivalent to 96 hours post inoculation) during infection. And authors measured its expression at much earlier time points in other experiments. Most importantly how authors justify its function as an effector, since this time point of 96 hpi is already in necrotrophic phase!
3. KO of MoSPAB1 experiments. Did authors measured the accumulated level of ROS in rice when inoculated with KO mutants of MoSPAB1?
4. Fig 3a and c look blurry and should be replaced. Authors better to have assays with sheath inoculation to quantitatively measure the level of invasive growth. And several places including Fig 3b did not have statistical data analysis. Plz justify why authors measured the expression much earlier time points and how about at 4 dpi?
5. It would be nice to see the data on binding of MoSPAB1 on the promoters of rice genes at genome scale. How many rice genes have the same binding sequence in the promoters?
6. Writing. Discussion should be restructured, shortened and rewritten. Most are the general and introductory information and have much redundancy.
7. Minor comments on terminology and misspellings, e.g. WT and MT competitor vs unlabeled and mutant probes...

Reviewer #3 (Remarks to the Author):

This manuscript discussed how fungal pathogens use secreted proteins to facilitate invasion. The authors discovered a specific fungal secreted protein called MoSPAB1 that directly binds to a gene promoter in the host, leading to increased expression of the gene and promoting disease. They also found similar proteins in other fungi that activate genes in different host plants. This is an interesting study which might reveal a conserved mechanism used by fungi to manipulate host genes and enhance their ability to cause disease. I would like the authors address some minor questions as below:

Both MoSPAB1 and MoSPAB2 can bind to the Bsr-d1 promoter, however, only MoSPAB1 can activate Bsr-d1 expression in the dual-luciferase assay. Is it because MoSPAB2 acts as a suppressor of the host plant Bsr-d1 expression?

Would overexpression of MoSPAB1 lead to enhanced disease symptoms?

How does MoSPAB1 enter the plant nucleus? Does MoSPAB1 contain any nuclei localisation signal?

The description for Figure 2d is not clear. Is the left panel for a fungus labelled with mCherry? It seems to be an infection at 36 hpi according to the text in line 178.

Line 178 - line 181, faint red light was observed at 36 hpi, and it is more visible in the nuclei and cytoplasm at 48 hpi. Is it because of the difference of the MoSPAB1 expression level? I don't see evidence from Figure 2a.

Figure 5, the authors proposed a working model for MoSPAB1 mediated pathogenicity demonstrated by the H₂O₂ degradation. There should be explanation in the main text with more details about H₂O₂ degradation and the relation with peroxidase genes.

Page 3, line 46-58. This paragraph used lots of space to explain RNA study, it can be deleted or rewritten. This manuscript focusing on secreted proteins and has nothing to do with mRNA study. The manuscript should be concise.

Point-by-point response to the reviewers

Reviewer #1 (Remarks to the Author):

General comment:

This study assessed the role of the MoSPAB1 protein of *M. oryzae* in the directly activation of rice Bsr-d1 expression to facilitate pathogenesis. The authors previously identified the rice protein Bsr-d1 that encodes a zinc-finger transcription factor that acts as a brake to slow down host defense response and prevent the host immune system from overrunning by increasing H₂O₂ degradation in plant cells. Through protein pulldowns (using the Bsr-d1 promoter as a bait) the author identified two secreted proteins (MoSPAB1 and MoSPAB2) that binds to Bsr-d1 promoter to facilitate its activation and Bsr-d1 gene expression. The author employed a broad range of molecular techniques to prove their hypothesis. Live cell imaging of fluorescently labeled MoSPAB1 proved its nuclear localization in rice and tobacco cells. Both deletion and overexpression of MoSPAB1 proved that this protein is involved in the virulence of *M. oryzae*. The data analysis and statistics presented in this manuscript are appropriate. However, the authors must show individual data points in all graphs. The manuscript is well written, and the outcomes should serve as references for fungal geneticists and plant pathologists. I support the publication of this manuscript in Nature Communications after proper corrections.

Response: We thank you for the high remark on our work. Following the suggestion “However, the authors must show individual data points in all graphs.”, we have made revision and now shown individual data points in all graphs.

I have some comments that the authors should address.

1. General Comment: I strongly suggest that the authors should use the term effector for the MoSPAB1 and MoSPAB2. If the author cannot classify these proteins as effectors, they need a strong justification stated in the manuscript.

Response: Thanks for the suggestion. We have now used the term “effector” for MoSPAB1 and MoSPAB2. (Line 1, Page 1; Line 116, Page 6)

Specific comments:

2. Abstract: Fungal pathogens typically use secreted proteins to suppress host immune activators to facilitate invasion. It should be written: secreted effector proteins

Response: We appreciate the suggestion and have changed “secreted proteins” to “secreted effector proteins” accordingly in the revision. (Line 24, Page 2)

3. Introduction: Line 71-72: The rice blast disease caused by the *Magnaporthe oryzae* fungus can lead to devastating yield losses worldwide. Please improve the sentence. Rice blast disease, caused by *Magnaporthe oryzae* (synonym of *Pyricularia oryzae* Sacc.), causes devastating yield losses worldwide.

Response: Following the suggestion, we have changed the sentence to “Rice blast disease, caused by *Magnaporthe oryzae* (synonym of *Pyricularia oryzae* Sacc.), leads to devastating yield losses worldwide”. (Lines 71-72, Page 4)

4. Figure 1: Authors must show individual data points in the graphs of this figure and in all graphs of the manuscript. Please present the graphs in color. Use color-blind friendly colors for the figures. Why the only one bar in Fig1B is in dark gray? Graph legends are touching the axes labels.

Response: Thanks for the suggestion. We have added individual data points in all graphs of the manuscript, and used color-blind friendly colors for the figures. Meanwhile, we have revised Fig1b accordingly in our revision.

5. Figure 2: Part of the scale bars in the protoplast pictures are outside of the panels. Please improve it. The legend mentions the yellow triangles. It is supposed to be written arrowheads that indicate the location of the BIC. Yellow triangles arrowheads represent indicates the biotrophic interfacial complex (BIC) of *M. oryzae*.

Response: We have made the scale bars inside of the panels of the protoplast pictures. Meanwhile, we have revised the sentence to “Yellow arrowheads indicate the biotrophic interfacial complex (BIC) of *M. oryzae*” in the annotation of Figure 2. (Line 794, Page 33)

6. Figure 3: Please present the graphs in color. Panel C must improve resolution.

Response: On the basis of this suggestion, we have presented figure 3 in color and improved resolution of Panel C in our revision.

7. Figure 4: The authors could make the circles around the lesion in Fig4F and G with a darker color. White is hard to see.

Response: We have made the circles around the lesions with a darker color in Figs 4f and 4g following the suggestion.

8. Figure 5: Legend: “During *M. oryzae* infection, MoSPAB1 was secreted from invasive hyphae to rice cytoplasm and the host nucleus”. It is missing the word “translocated” It should be written: “During *M. oryzae* infection, MoSPAB1 was secreted from invasive hyphae and translocated to rice cytoplasm and the host nucleus.

Response: We have revised the legend of Figure 5 following the suggestion by adding “and translocated”. (Lines 866-867, Page 36)

9. References: Please double check journal abbreviations.

Response: We have checked the journal abbreviations again and revised them. (Lines 649, 674-675, and 691, Pages 29-30)

10. Supplemental Figures 9 and 10. The figures have poor resolution. Dendrograms and alignments should be presented in color.

Response: We have replaced the figures with those of higher resolution for Supplemental Figures 9 and 10 and have remade the dendrograms and alignments in color.

Reviewer #2 (Remarks to the Author):

The authors report a novel mechanism for fungal infection in plant using the model pathosystem, rice and *Magnaporthe oryzae* interactions. The fungal effector MoSPAB1 secreted by the blast fungus binds to the promotor of host gene, *Bsr-d1*, and activates its expression. Higher expression of *Bsr-d1* makes rice plant more susceptible against the blast fungal infection. Furthermore, authors found MoSPAB1 and host MYBS1 compete to bind to the promotor of *Bsr-d1*. Experiments were straight forward and performed scientifically, but the data and figures presented in the current form need to be upgraded or replaced. And Discussion should be shortened and restructured. Followings are the major and minor comments to improve the manuscript.

Response: Thanks for these suggestions. We have upgraded or replaced the data and figures (Figures 2, 3, S4, S9, and S11). We also have shortened and restructured the discussion (Lines 301-392, Pages 15-19).

1. Localization of MoSPAB1. Fig 2D is not clear to see the localization of MoSPAB1. It is very difficult to figure out where is the appressorium, BIC and rice

nucleus and whether they are in invaded or neighboring cells. It should be upgraded or replaced! And what is the ratio of cytoplasm and nucleus localization of MoSPAB1?

Response: Thanks for these suggestions. We have replaced the figures with those of higher resolution and it is now clear to see the localization of MoSPAB1 in Figure 2d. We have calculated the ratio of cytoplasmic MoSPAB1 to nuclear MoSPAB1 following the suggestion. Its ratio is about 2:1 (Lines 189-190, Page 9). These data are shown in Figure 2e in our revision.

2. Expression of MoSPAB1. Its expression was highest at 4 dpi (equivalent to 96 hours post inoculation) during infection. And authors measured its expression at much earlier time points in other experiments. Most importantly how authors justify its function as an effector, since this time point of 96 hpi is already in necrotrophic phase!

Response: MoSPAB1 possesses a signal peptide and can enter rice cells. Meanwhile, MoSPAB1 targets and regulates *Bsr-d1* in rice cells to facilitate invasion. We believe these results justify our conclusion that MoSPAB1 is an effector.

We have repeatedly measured *MoSPAB1* expression. Our results still show that *MoSPAB1* RNA expression level is highest at 4 dpi (96 hpi). *MoSPAB1* RNA level was particularly high at infection stage (Lines 165-167, Page 8). The data were shown in the new Figure 2a.

It is necrotrophic phase when *M. oryzae* infection is at 96 hpi. However, the necrotrophic phase is mainly present in the original infection site on rice. The areas surrounding the necrotrophic-phase site are still at biotrophic phase. Therefore, at 96 hpi, MoSPAB1 still has a large effect as a *M. oryzae* effector. Similarly, Chen et al. (2013) showed that many effectors secreted from *M. oryzae* are detected at 96 and 120 hpi (“Identification and characterization of in planta-expressed secreted effector proteins from *Magnaporthe oryzae* that induce cell death in rice” in *Mol. Plant-Microbe Interact.*) This reference supports our results that *MoSPAB1* expression level is high at 96 hpi.

3. KO of MoSPAB1 experiments. Did authors measured the accumulated level of ROS in rice when inoculated with KO mutants of MoSPAB1?

Response: We have added these data in Figure 3b in our revision. We have detected H₂O₂ level in rice when inoculated with wild-type Guy11, a *MoSPAB1* deletion mutant (Δ *Mospab1-1*) and its complemented strain. Our result indicates that Δ *Mospab1* induces more ROS than wild-type Guy11 and the complemented strain (Lines 204-205, Page 10). The data are shown in the new Figure 3b.

4. Fig 3a and c look blurry and should be replaced. Authors better to have assays with sheath inoculation to quantitatively measure the level of invasive growth. And several places including Fig 3b did not have statistical data analysis. Plz justify why authors measured the expression much earlier time points and how about at 4 dpi?

Response: Thanks for these suggestions. Figs 3a and 3c have been replaced with high resolution figures. Fig 3c is now Fig 3d because a new subfigure is added. The experiments of sheath inoculation have been added accordingly. The data are shown in the new Supplementary Figure 7. Additionally, statistical data analysis has been carried out for Fig3b and other figures.

Bsr-dl can be induced by *M. oryzae* at earlier time points. Therefore, we measured the expression of *Bsr-dl* at earlier time points. Based on your suggestion, we also measured *Bsr-dl* levels at 4 dpi (96 hpi). The pattern of *Bsr-dl* expression for Guy11, Δ *Mospab1*, and complemented strain at 4 dpi are similar to those at earlier time points. The data were shown in Figure 3c in our revision.

5. It would be nice to see the data on binding of MoSPAB1 on the promoters of rice genes at genome scale. How many rice genes have the same binding sequence in the promoters?

Response: We appreciate the suggestion and have analyzed the motif in the rice genome. The motif sequence bound by a transcription factor is widely present in genome. However, this transcription factor may not regulate all genes with the conserved motif in their promoters. We found 8672 genes (not including *Bsr-dl*) in the rice genome that carry at least one (C/A)(G/C)(C/G)T(T/C)G(C/A)(T/C) motif in their promoters (-1500 bp). Among these genes, we found 127 of them (not including *Bsr-dl*) carry three or more tandem conserved motifs in their promoters (-1500 bp). However, the average distance of the tandem motifs in *Bsr-dl* promoter was the shortest among these genes. It indicates that *Bsr-dl* is regulated by MoSPAB1 with the strongest regulatory ability.

6. Writing. Discussion should be restructured, shortened and rewritten. Most are the general and introductory information and have much redundancy.

Response: Thanks for this suggestion. We have restructured and shortened the discussion. (Lines 301-392, Pages 15-19)

7. Minor comments on terminology and misspellings, e.g. WT and MT competitor vs unlabeled and mutant probes... (Lines 143, 148)

Response: Thanks for the suggestion. We have corrected these. The sentence “we included wild type and mutated competitors” is changed to “we include unlabeled and mutant competitors”. And “while mutated competitors had little effects” is changed to “while mutant competitors had little effects”. (Lines 145-146, and 150, Page 7)

Reviewer #3 (Remarks to the Author):

This manuscript discussed how fungal pathogens use secreted proteins to facilitate invasion. The authors discovered a specific fungal secreted protein called MoSPAB1 that directly binds to a gene promoter in the host, leading to increased expression of the gene and promoting disease. They also found similar proteins in other fungi that activate genes in different host plants. This is an interesting study which might reveal a conserved mechanism used by fungi to manipulate host genes and enhance their ability to cause disease.

Response: We thank you for the high evaluation on our work.

I would like the authors address some minor questions as below:

1. Both MoSPAB1 and MoSPAB2 can bind to the *Bsr-d1* promoter, however, only MoSPAB1 can activate *Bsr-d1* expression in the dual-luciferase assay. Is it because MoSPAB2 acts as a suppressor of the host plant *Bsr-d1* expression?

Response: Our result suggests that both MoSPAB1 and MoSPAB2 can bind to the *Bsr-d1* promoter. But MoSPAB2 may not act as a suppressor or activator to regulate *Bsr-d1* based on the data that MoSPAB2 does not affect *Bsr-d1* expression in the dual-luciferase assay (Supplementary Figure 1c). This is actually a very common phenomenon. Many transcription factors can bind to the same binding site, but usually only few can activate the gene containing the binding site.

2 Would overexpression of MoSPAB1 lead to enhanced disease symptoms?

Response: Yes. We generated transgenic rice overexpressing *MoSPAB1* and found that *MoSPAB1* overexpression in rice leads to enhanced blast disease symptoms (Lines 212-215, Pages 10-11). Please find the result in Figure 3d.

3. How does MoSPAB1 enter the plant nucleus? Does MoSPAB1 contain any nuclei localisation signal?

Response: MoSPAB1 might use unknown nuclear localization signals to enter plant nucleus. However, we did not find classical nuclear localization signals in the MoSPAB1 protein. We found four regions with at least two K or R amino acids that could potentially serve as nuclear localization signals (New Supplementary Fig 3a). However, when we mutated each region they did not change MoSPAB1 nuclear localization (New Supplementary Fig 3b). This indicates that MoSPAB1 might use non-classical nuclear localization signals or might be facilitated by other factors to enter host nuclei (Lines 175-182, Page 9).

4. The description for Figure 2d is not clear. Is the left panel for a fungus labelled with mCherry? It seems to be an infection at 36 hpi according to the text in line 178.

Response: Thanks for this kind reminder. We have added this information to Figure 2d legend to clarify it (Lines 791-794, Page 33). The upper panel is for fungal cells labelled with mCherry. We used the *MoSPAB1* promoter to drive mCherry expression. And it is in fact an infection at 36 hpi (Supplementary Fig 4).

5 Line 178 - line 181, faint red light was observed at 36 hpi, and it is more visible in the nuclei and cytoplasm at 48 hpi. Is it because of the difference of the MoSPAB1 expression level? I don't see evidence from Figure 2a.

Response: Thanks for the comment. We have detected *MoSPAB1* RNA levels at 36 and 48 hpi in our revision. The new result shows that *MoSPAB1* level is higher at 48 hpi than at 36 hpi. These data are shown in Figure 2a.

6. Figure 5, the authors proposed a working model for MoSPAB1 mediated pathogenicity demonstrated by the H₂O₂ degradation. There should be explanation in the main text with more details about H₂O₂ degradation and the relation with peroxidase genes.

Response: Thanks for the suggestion. We have added a brief description about H₂O₂ degradation and the relation with peroxidase genes in the main text. (Line 308-311, Page 15). For details about Bsr-d1-mediated H₂O₂ degradation and associated peroxidases, please refer to our previous report (Li, et al. A natural allele of a transcription factor in rice confers broad-spectrum blast resistance. 2017, *Cell*. **170**, 114-126)

7. Page 3, line 46-58. This paragraph used lots of space to explain RNA study, it can be deleted or rewritten. This manuscript focusing on secreted proteins and has nothing to do with mRNA study. The manuscript should be concise.

Response: Thanks for this suggestion. We have revised it accordingly. We have deleted the part of mRNA study and kept the references of secreted proteins in the Introduction. (Lines 50-54, Page 3)

REVIEWER COMMENTS

Reviewer #1 (Remarks to the Author):

The author incorporated most of the modifications requested by the reviewers. The Manuscript is very well written and suitable for publication in Nature Communications.

Reviewer #3 (Remarks to the Author):

The concerns I had with the first submission have been well addressed by the authors.

Cross-comment:

I've looked at the responses to Reviewer 2. The authors addressed most of the questions from Reviewer 2 very well. However, some of the questions are not. Please find details below:

Reviewer 2 asked the authors to provide better resolution figure for Fig 2D. Fig 2D is still very blurry in this revision version.

Reviewer 2 also asked whether MoSPAB1 signals are in the invaded cell or also in the neighbouring cells. The authors didn't respond to this question.

Point-by-point response to the reviewers

Reviewer #1 (Remarks to the Author):

The author incorporated most of the modifications requested by the reviewers. The Manuscript is very well written and suitable for publication in Nature Communications.

Response: We thank you for the high evaluation on our work.

Reviewer #3 (Remarks to the Author):

The concerns I had with the first submission have been well addressed by the authors.

Response: We thank you find these changes satisfactory and the high remark on our work.

Cross-comment:

I've looked at the responses to Reviewer 2. The authors addressed most of the questions from Reviewer 2 very well. However, some of the questions are not. Please find details below:

(1) Reviewer 2 asked the authors to provide better resolution figure for Fig 2D. Fig 2D is still very blurry in this revision version.

Response: We have replaced the figure 2d with higher resolution. It is now clear to see the localization of MoSPAB1 in Figure 2d. We also use yellow triangles to represent the appressorium in Figure 2d, and note it in the legend of Figure 2 (Lines 764-765, Page 32).

(2) Reviewer 2 also asked whether MoSPAB1 signals are in the invaded cell or also in the neighbouring cells. The authors didn't respond to this question.

Response: Figure 2d shows that MoSPAB1 signals (red signals) are in the invaded cell. However, we did not find red signals in its neighboring cells in Figure 2d, which suggests that MoSPAB1 signals are not in the neighboring cells of the invaded cell.